# *Enhancing Code-Switching for Cross-lingual SLU*: A Unified View of Semantic and Grammatical Coherence

**Zhihong Zhu**[†]   **Xuxin Cheng**[†]   **Zhiqi Huang**
**Dongsheng Chen**   **Yuexian Zou**[*]
School of ECE, Peking University, China
{zhihongzhu, chengxx, chends}@stu.pku.edu.cn
{zhiqihuang, zouyx}@pku.edu.cn

## Abstract

Despite the success of spoken language understanding (SLU) in high-resource languages, achieving similar performance in low-resource settings, such as zero-shot scenarios, remains challenging due to limited labeled training data. To improve zero-shot cross-lingual SLU, recent studies have explored code-switched sentences containing tokens from multiple languages. However, vanilla code-switched sentences often lack semantic and grammatical coherence. We ascribe this lack to two issues: (1) randomly replacing code-switched tokens with equal probability and (2) disregarding token-level dependency within each language. To tackle these issues, in this paper, we propose a novel method termed SoGo, for zero-shot cross-lingual SLU. First, we use a saliency-based substitution approach to extract keywords as substitution options. Then, we introduce a novel token-level alignment strategy that considers the similarity between the context and the code-switched tokens, ensuring grammatical coherence in code-switched sentences. Extensive experiments and analyses demonstrate the superior performance of SoGo across nine languages on MultiATIS++.

## 1 Introduction

Acting as the interface between users and machines, spoken language understanding (SLU) is crucial in task-oriented dialogue systems (Qin et al., 2021; Chen et al., 2022; Zhu et al., 2023b; Cheng et al., 2023c). While joint training models have led to significant advancements, most existing models rely on annotated training data, limiting their scalability to low-resource languages. To this end, zero-shot cross-lingual SLU gains increasing attention.

Existing works have shown the effectiveness of Multilingual BERT (mBERT) in multilingual corpus pre-training, enabling promising zero-shot cross-lingual SLU (Deshpande et al., 2022; Cheng et al., 2023b; Zhu et al., 2023a). Therein, Qin et al. (2020) extended this concept to a code-switched setting, aligning the source language with multiple target languages. Qin et al. (2022) incorporate contrastive learning to achieve fine-grained cross-lingual transfer. Based on this, Liang et al. (2022) further propose a multi-level contrastive learning framework for explicit alignment of utterance-slot-word structure in cross-lingual SLU.

However, most above-mentioned models adopt code-switching methods, which often lack both semantic coherence and grammatical coherence. This can be attributed to two main issues: (1) Current approaches overlook the importance of individual words in a sentence, as they are randomly replaced with equal probability, potentially introducing unnecessary translation burden and noise that affects semantic coherence. (2) Code-switched sentences comprising tokens from multiple languages may lack grammatical coherence, as token-level dependencies within each language are disregarded.

To address these issues, we propose a novel framework dubbed SoGo for enhancing code-switching. Concretely, we first employ a saliency-based substitution approach to identify keywords with high saliency scores. These keywords are then replaced with their target language equivalents using bilingual dictionaries, to generate code-switched sentences. Notably, unlike the approach by Liu et al. (2020), our SoGo framework requires only a single training process for multiple target languages. Then, we devise a novel token-level alignment strategy to bridge the gap between original context and code-switched tokens. By mapping code-switched tokens back to the original context, SoGo minimizes the impact of token substitutions on token-level coherence, thus maintaining consistent contextual space within each language.

Experimental results on MultiATIS++ (Xu et al., 2020) demonstrate that employing code-switching

---

[†]Equal contribution.
[*]Corresponding author.

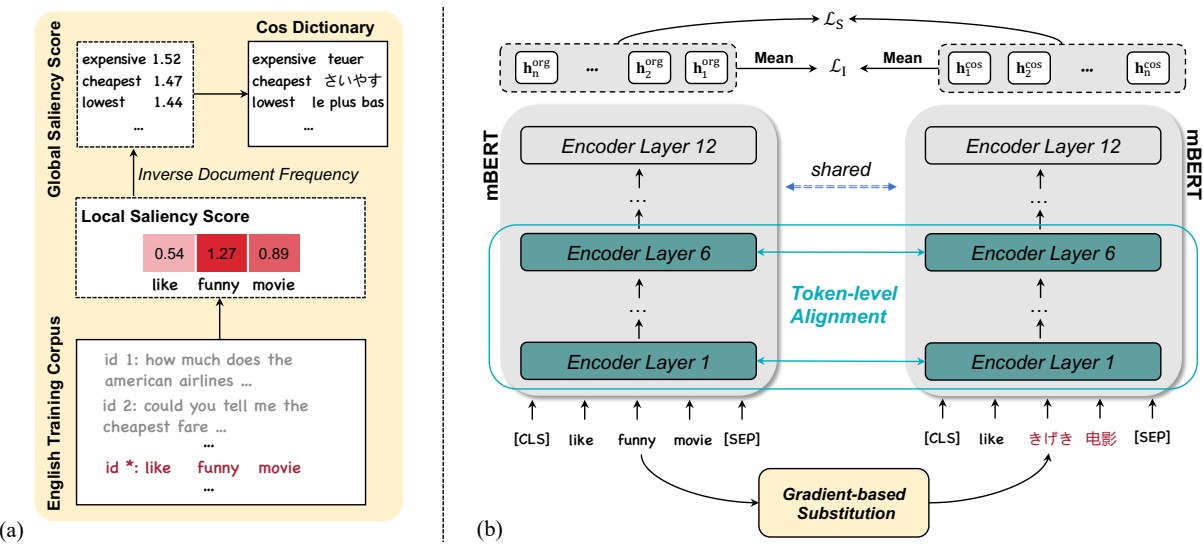

Figure 1: (a) **Our proposed gradient-based substitution in SOGO**. The code-switched dictionary comprised of salient words has the capacity to more accurately characterize the common semantic space and facilitate the acquisition of semantic connections among different languages by the model. (b) **The main architecture of SOGO**. We perform token-level alignment in lower six layers of mBERT to enhance the grammatical coherence of the code-switched sentence.

with SOGO leads to significant performance improvement compared to baseline models. Furthermore, we demonstrate the compatibility of SOGO with other existing SLU structures.

## 2 Method

We describe problem formulation (§2.1) and backbone model (§2.2) for zero-shot cross-lingual SLU first, before describing SOGO (§2.3). The main architecture of SOGO is illustrated in Figure 1.

### 2.1 Problem Formulation

Given each input sentence $x_{tgt}$ in a target language, zero-shot cross-lingual SLU means the SLU model is trained in a source language dataset $S$, *e.g.*, {English}, and directly applied to other target languages datasets $T$, *e.g.*, {Chinese, Japanese}:

$$(o_{tgt}^I, o_{tgt}^S) = f(x_{tgt}),  \quad (1)$$

where $f(\cdot)$ is the trained model. $o_{tgt}^I$ represents an intent label and $o_{tgt}^S$ represents a slot sequence. Note that $T$ may consist of multiple $tgt$ languages.

### 2.2 Backbone Model

For each input sentence $x$, we follow Qin et al. (2022); Liang et al. (2022) to utilize bilingual dictionaries (Lample et al., 2018) to generate its corresponding multi-lingual one $x'$. Following previous studies, we select mBERT (Devlin et al., 2019)

as our sentence encoder. Formally, the input of mBERT is formulated as follows:

$$x' = ([\text{CLS}], x_1', \ldots, x_n', [\text{SEP}]),  \quad (2)$$

where [CLS] denotes the special symbol for representing the whole sequence, and [SEP] can be used for separating non-consecutive token sequences (Devlin et al., 2019).

**Intent Detection.** We feed the sentence representation $h'_{\text{CLS}}$ generated from mBERT into a classification layer to obtain $o^I : o^I = \text{softmax}(W^I h'_{\text{CLS}} + b^I)$, where $W^I$ and $b^I$ are tunable parameters.

**Slot Filling.** We use the hidden state to predict each slot: $o_t^S = \text{softmax}(W^S h_t' + b^S)$, where $h_t'$ denotes the representation of word $x_t'$. Note that mBERT tokenizes words into subwords (Wang et al., 2019), so we adopt the first sub-token's representation as the whole word representation.

**Training and Inference.** Following previous works, the loss function for intent detection ($\mathcal{L}_I$) and slot filling ($\mathcal{L}_S$) are optimized jointly by corresponding cross-entropy loss. Finally, the total training loss is the weighted sum of two losses:

$$\mathcal{L} = \lambda_I \mathcal{L}_I + \lambda_S \mathcal{L}_S,  \quad (3)$$

where $\lambda_I$ and $\lambda_S$ denote two hyper-parameters.

During inference, we do not use any code-switched sentences.

### 2.3 SoGo

**Saliency-based Substitution.** Intuitively, each word in a sentence contributes differently, and we refer to words with significant impact as keywords. Given a vocabulary set $\mathcal{V}$ containing $v$ words, we can identify a salient subset of keywords $\mathcal{K} \subseteq \mathcal{V}$ to generate code-switched sentences. In our work, gradient-based saliency scores (Li and Yu, 2015; Denil et al., 2014; Arras et al., 2019) are used to select keywords, which has been proven effective in many other tasks (Yang et al., 2022; Lai et al., 2021; Lei et al., 2023). Methodologically, let $x_n$ denotes the $n$-th sentence from $S$, $\mathcal{L}_{\hat{y}_n}$ is the loss between model's prediction of slot filling $\hat{y}_n$ and the golden label $y_n$. For each token $w_{i,n} \in x_n$, the saliency score is defined as:

$$S_{x_n}(w_{i,n}) = -\nabla_{h_{i,n}} \mathcal{L}_{\hat{y}_n} \times h_{i,n}, \qquad (4)$$

where $h_{i,n}$ represents the embedding of $w_{i,n}$. Note that mBERT tokenizes words into subwords, so we calculate the average of the subword saliency scores for each word to obtain the final score. In this manner, the *gradient* signifies the degree to which a word embedding influences the final decision, while the *input* considers both the sign and magnitude of the input (Shrikumar et al., 2017).

Equation 4 assesses the local significance of a token within a sentence. However, our objective is to contrast a global keyword subset $\mathcal{K}$ from $S$. Following Yuan et al. (2020), we aggregate all saliency scores for the token $w_i$ occurring in $S$. We then multiply these scores with the inverse document frequency (IDF) (Robertson, 2004) of the token $w_i$:

$$S(w_i) = \log \frac{N}{|\{x \in S : w_i \in x\}|} \cdot \sum_{x \in S : w_i \in x} S_x(w_i), \qquad (5)$$

in which $N$ is the total number of words in $S$. The IDF term balances word frequency and saliency scores by assigning words with high document frequency a lower weight and vice versa.

Finally, top-$k$ salient words are chosen to compose the code-switched dictionary $\mathcal{K}$. We opt for the dictionary $\mathcal{K}$ as a replacement strategy instead of random word substitution in sentences.

**Token-level Alignment.** To further align the space of all hidden states in the code-switched sentence, inspired by attention-based alignment methods Zhu et al. (2023c); Feng et al. (2022), we map the code-switched sentence to the original sentence by calculating the similarity between substituted hidden states and original hidden states. The similarity scores are then used as weights to aggregate the embedding of original hidden states, reflecting the calculated potential for substituted hidden states in the code-switched sequence. To be specific, for each hidden state $h_i'$ in $h'$, the similarity score is calculated as Vaswani et al. (2017):

$$\text{score}_i' = h_i' \cdot h^\top. \qquad (6)$$

The final potential of the code-switched token $\tilde{h}_i'$ is the weighted sum of its corresponding $h$:

$$\alpha_{i,t} = \frac{\exp(\text{score}_{i,t}')}{\sum_{t'} \exp(\text{score}_{i,t'}')}, \quad \tilde{h}_i' = \sum_t \alpha_{i,t} \cdot h_t. \qquad (7)$$

Note that we only employ the alignment in lower six layers to better align the representations for switched tokens due to their superior syntax-capturing ability (Jawahar et al., 2019; Rogers et al., 2020). Eventually, the loss function for each sub-task $\mathcal{L}_*$ ($* \in [I, S]$) in Equation 3 is rewritten as:

$$\mathcal{L}_* = \frac{\mathcal{L}(h'^*, y^*) + \mathcal{L}(h^*, y^*)}{2}, \qquad (8)$$

in which $y^*$ denotes golden label of each task. When $*$ denotes $I$, we employ mean pooling strategy on representations of all tokens instead of using $h^{\text{CLS}}$ and $h'^{\text{CLS}}$ for intent detection, considering the close relationship of the two tasks.

## 3 Experiments

### 3.1 Experimental Setting

**Dataset and Metrics.** We conduct our experiments on the latest multilingual benchmark dataset MultiATIS++ (Xu et al., 2020),[1] which consists of nine languages: English (en), Spanish (es), Portuguese (pt), German (de), French (fr), Chinese (zh), Japanese (ja), Hindi (hi), and Turkish (tr). The details of MultiATIS++ are shown in Appendix A.

Following Li et al. (2021), we evaluate the performance of intent prediction using accuracy, slot filling using F1 score, and the sentence-level semantic frame parsing using overall accuracy.

**Implementation Details.** We adopt two baseline models as the backbone, *i.e.*, CoSDA (Qin et al., 2020) and GL-CLeF (Qin et al., 2022). Following previous works, the random rate $\alpha$ for token substitution is set to 0.2. AdamW (Loshchilov and

---

[1] https://github.com/amazon-science/multiatis

| | Model | en | de | es | fr | hi | ja | pt | tr | zh | Avg. |
|---|---|---|---|---|---|---|---|---|---|---|---|
| **Intent Acc** | mBERT♡ (Devlin et al., 2019) | 98.54 | 95.40 | 96.30 | 94.31 | 82.41 | 76.18 | 94.95 | 75.10 | 82.53 | 88.42 |
| | ZSJoint◇ (Chen et al., 2019) | 98.54 | 90.48 | 93.28 | 94.51 | 77.15 | 76.59 | 94.62 | 73.29 | 84.55 | 87.00 |
| | CoSDA♡ (Qin et al., 2020) | 95.74 | 94.06 | 92.29 | 77.04 | 82.75 | 73.25 | 93.05 | 80.42 | 78.95 | 87.32 |
| | SoGo $_{CoS}$ (Ours) | 98.54 | 96.97 | 98.12 | 96.33 | 82.76 | 76.41 | 97.18 | 82.96 | 85.77 | 90.56 |
| | GL-CLEF♡ (Qin et al., 2022) | 98.77 | 97.53 | 97.05 | 97.72 | **86.00** | 82.84 | 96.08 | 83.92 | 87.68 | 91.95 |
| | SoGo $_{GL}$ (Ours) | **98.89** | **98.45** | **98.15** | 97.74 | 83.87 | **84.75** | **97.73** | **85.53** | **89.10** | **92.69** |
| | LAJ-MCL◇ (Liang et al., 2022) | 98.77 | 98.10 | 98.10 | **98.77** | 84.54 | 81.86 | 97.09 | 85.45 | 89.03 | 92.41 |
| **Slot F1** | mBERT♡ (Devlin et al., 2019) | 95.11 | 80.11 | 78.22 | 82.25 | 26.71 | 25.40 | 72.37 | 41.49 | 53.22 | 61.66 |
| | ZSJoint◇ (Chen et al., 2019) | 95.20 | 74.79 | 76.52 | 74.25 | 52.73 | 70.10 | 72.56 | 29.66 | 66.91 | 68.08 |
| | CoSDA♡ (Qin et al., 2020) | 92.29 | 81.37 | 76.94 | 79.36 | 64.06 | 66.62 | 75.05 | 48.77 | 77.32 | 73.47 |
| | SoGo $_{CoS}$ (Ours) | 95.46 | 84.12 | 83.84 | 83.46 | 57.63 | 65.78 | 80.27 | 55.13 | 79.56 | 76.14 |
| | GL-CLEF♡ (Qin et al., 2022) | 95.39 | 86.30 | 85.22 | 84.31 | 70.34 | 73.12 | 81.83 | 65.85 | 77.61 | 80.00 |
| | SoGo $_{GL}$ (Ours) | 95.42 | **87.46** | **87.01** | **84.45** | **74.25** | **76.69** | **83.91** | **67.04** | 78.53 | **81.64** |
| | LAJ-MCL◇ (Liang et al., 2022) | **96.02** | 86.59 | 83.03 | 82.11 | 61.04 | 68.52 | 81.49 | 65.20 | **82.00** | 78.23 |
| **Overall Acc** | mBERT♡ (Devlin et al., 2019) | 87.12 | 52.69 | 52.02 | 37.29 | 4.92 | 7.11 | 43.49 | 4.33 | 18.58 | 36.29 |
| | ZSJoint◇ (Chen et al., 2019) | 87.23 | 41.43 | 44.46 | 43.67 | 16.01 | 33.59 | 43.90 | 1.12 | 30.80 | 38.02 |
| | CoSDA♡ (Qin et al., 2020) | 77.04 | 57.06 | 46.62 | 50.06 | 26.20 | 28.89 | 48.77 | 15.24 | 46.36 | 44.03 |
| | SoGo $_{CoS}$ (Ours) | 88.35 | 61.43 | 58.30 | 56.37 | 19.70 | 27.65 | 60.43 | 19.30 | 50.17 | 49.08 |
| | GL-CLEF♡ (Qin et al., 2022) | 88.02 | 66.03 | 59.53 | 57.02 | 34.83 | 41.42 | 60.43 | 28.95 | 50.62 | 54.09 |
| | SoGo $_{GL}$ (Ours) | **90.54** | **72.26** | **61.05** | **57.88** | **39.90** | **46.95** | **64.23** | **29.14** | 51.31 | **57.02** |
| | LAJ-MCL◇ (Liang et al., 2022) | 89.81 | 67.75 | 59.13 | 57.56 | 23.29 | 29.34 | 61.93 | 28.95 | **54.76** | 52.50 |

Table 1: Major results (%) on MultiATIS++. Higher is better in all columns. Results with ♡ are taken from Qin et al. (2022), and results with ◇ are taken from Liang et al. (2022). SoGo $_{CoS}$ and SoGo $_{GL}$ can be directly compared to CoSDA and GL-CLEF. Note that LAJ-MCL is not open-source.

Hutter, 2019) is used to train SoGo with a learning rate of 5$e$-6. The coefficients in Equation 3 are $\lambda_I$ and $\lambda_S$, where the best ones are selected by searching a combination with the following ranges: $\{0.8, 0.9, 1.0, 1.1, 1.2\}$. All experiments are conducted on a single TESLA-V100. The experimental results of our method are averaged over five runs with different random seed to ensure the final reported results are statistically stable.

### 3.2 Major Results

From the major results in Table 1, we have the following observations: (1) SoGo $_{CoS}$ outperforms CoSDA significantly. This is because that code-switched sentences generated by performing salient word substitution and token-level alignment maintain better syntactic and semantic consistency than those generated by random substitution. (2) Both SoGo $_{CoS}$ and SoGo $_{GL}$ beat their counterparts with 5.05% and 2.93% average improvement in overall accuracy, respectively. This demonstrates that SoGo can be integrated into existing SLU models, orthogonal of other structures. Notably, the gain in slot filling is more significant than in intent detection, further validating the effectiveness of token-level alignment. (3) As LAJ-MCL is not open-source, we implemented two versions of

SoGo, referred to as SoGo $_{CoS}$ and SoGo $_{GL}$, to compare with LAJ-MCL. Remarkably, SoGo $_{GL}$ outperformed the competitive LAJ-MCL in overall accuracy across 8/9 languages. This highlights the significant potential of our SoGo when combined with existing SLU models.

### 3.3 Further Analysis

**Ablation Study.** We conduct an ablation study to explore the effect of each component in SoGo. The results are illustrated in Figure 2, which show that each proposed component (*i.e.*, saliency-based substitution and token-level alignment) contributes to the performance positively. It is obvious that the proposed saliency-based substitution has a strong ability to pick out important words to boost performance. Moreover, our token-level alignment has shown greater relative importance of the switched tokens than the vanilla code-switching method. It indicates that such substituted tokens significantly contribute to SoGo's prediction.

**Analysis of Saliency-based Substitution.** To explore the potential of saliency-based substitution, we perform experiments on SoGo $_{CoS}$ of different strategies with respect to different keyword replacement rate $\frac{k}{v}$ (*cf.* Figure 3, where $k$ denotes

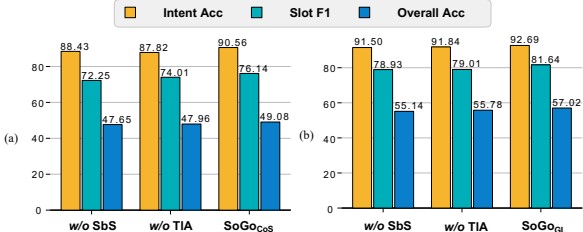

Figure 2: Ablation experiments. "SbS" and "TlA" denote saliency-based substitution and token-level alignment, respectively.

the number of keywords, and $v$ denotes the total number of words.). We find the performance of randomly selecting keywords significantly declines as $\frac{k}{v}$ decreases, whereas our saliency-based substitution continues to perform well even with only 1% of keywords. This is because saliency-based substitution prioritizes the most indicative keywords for code-switching. As $\frac{k}{v}$ increases, the additional keywords become less indicative and might even have a negative effect on the model's performance.

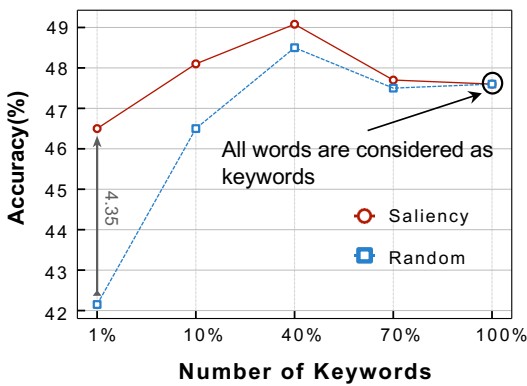

Figure 3: Overall Accuracy of SoGo $_{CoS}$ in different replacement rate of keywords $\frac{k}{v}$. Random denotes selecting keywords randomly, while `Saliency` denotes selecting keywords via saliency-based substitution.

**Analysis of Token-level Alignment.** To verify whether token-level alignment preserves a context space to keep grammatical coherence, we plot the degree of dispersion among tokens within a random sentence in Figure 4. Concretely, we retrieve the token embedding from intermediate (sixth) layers for both switched and original tokens and utilize top features calculated from PCA (Abdi and Williams, 2010). From Figure 4, we can conclude that SoGo $_{CoS}$ has shown a more compact space for the original tokens, where switched tokens are also inside the context space. In contrast, the vanilla code-

switching method entirely separates the substituted tokens apart from the original context.

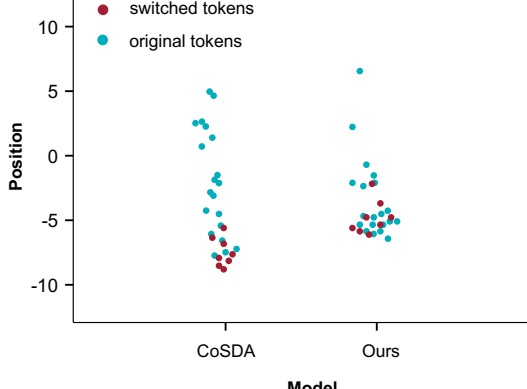

Figure 4: Token-Level density of SoGo $_{CoS}$ during training procedure.

## 4 Conclusion

We proposed a novel s̲emantics-c̲oherent and g̲rammar-c̲oherent method (SoGo) to enhance code-switching for zero-shot cross-lingual SLU. Specifically, we utilized saliency-based substitution to replace keywords for better semantic coherence in the shared space. Besides, we introduced an alignment strategy to map the code-switched tokens to original contexts, which solves the grammatical incoherence of code-switching. Experiments on MultiATIS++ showed that code-switching based methods with SoGo obtained competitive performance compared to counterpart baselines.

## Limitations

By applying saliency-based substitution and token-level alignment, SoGo achieves significant improvement on the benchmark datasets. Nevertheless, we summarize two limitations for further discussion and investigation by other researchers:

(1) Like other code-switching methods, the performance of the model still relies heavily on the correctness of the bilingual dictionary (Fazili and Jyothi, 2022; Whitehouse et al., 2022). In the future, we continue to enhance the performance of zero-shot cross-lingual SLU by refining code-switching techniques (Cheng et al., 2023a).

(2) SoGo should further leverage language-independent information (Yu et al., 2021), such as dependency relations and POS tags, to ensure syntactic invariance for cross-lingual transfers.

## Acknowledgements

We thank all anonymous reviewers for their constructive comments. This paper was partially supported by Shenzhen Science & Technology Research Program (No:GXWD20201231165807007-20200814115301001) and NSFC (No: 62176008).

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

## A  Dataset

MixATIS++ is an extension of Multilingual ATIS (Table 2). Human-translated data for six languages including Spanish (es), German (de), Chinese (zh), Japanese (ja), Portuguese (pt), French (fr) are added to Multilingual ATIS which initially has Hindi (hi) and Turkish (tr). There are 4,478 utterances in the train set, 500 in the valid set, and 893 in the test set, with 18 intents and 84 slots for each language.

| Language | Utterances | | | Intent types | Slot types |
|---|---|---|---|---|---|
| | #Train | #Valid | #Test | | |
| hi | 1,440 | 160 | 893 | 17 | 75 |
| tr | 578 | 60 | 715 | 17 | 71 |
| others | 4,488 | 490 | 893 | 18 | 84 |

Table 2: Statistics of MultiATIS++.

## B  Experiment Details

Following the zero-shot setting, we fine-tune the model on en training set and use en validation set for the hyper-parameters search. The best model checkpoint is decided by the overall accuracy on en validation set. Our code is based on PyTorch and Transformers[2].

## C  Baselines

We compare our model to the following baselines.

**mBERT.** mBERT[3] follows the same model architecture as BERT (Devlin et al., 2019), but instead of training only on monolingual English data, it is trained on the Wikipedia pages of 104 languages with a shared word piece vocabulary, allowing the model to share embeddings across languages.

**ZSJoint.** Chen et al. (2019) propose a zero-shot SLU model, which is trained on the en training set and directly applied to the test sets of target languages.

**SoSDA.** Qin et al. (2020) propose a data augmentation framework to generate multi-lingual code-switching data to fine-tune mBERT, which encourages the model to align representations from the source and multiple target languages.

---

[2]https://github.com/huggingface/transformers
[3]https://github.com/google-research/bert/blob/master/multilingual.md

**GL-CLEF.** Qin et al. (2022) introduce a contrastive learning framework to explicitly align representations across languages for zero-shot cross-lingual SLU.

**LAJ-MCL.** Liang et al. (2022) proposes a multi-level contrastive learning framework for zero-shot cross-lingual SLU.

## D  Generalizability of SoGo

We investigate the potential of the proposed model for other tasks through an additional set of experiments. Following Qin et al. (2020), we conduct additional experiments for natural language inference using XNLI (Conneau et al., 2018), which includes 15 languages. We directly input pairs of sentences into the mBERT encoder, and a task-specific classification layer is employed for classification. The models are evaluated based on classification accuracy (ACC), and the results are presented in Table 3.

| Model | Avg. |
|---|---|
| CoSDA_XLM-based (Conneau and Lample, 2019) | 75.3 |
| + SoGo (Ours) | 77.4 |
| CoSDA_mBERT-based (Wu and Dredze, 2019) | 69.7 |
| + SoGo (Ours) | 71.9 |

Table 3: Natural language inference experiments.