# OpenReview forum: "Enhancing Code-Switching for Cross-lingual SLU: A Unified View of Semantic and Grammatical Coherence"
_EMNLP/2023/Conference — EMNLP 2023 Main_

### Official Review · Reviewer_nHnV · 2023-08-05

**Soundness:** 4

**Excitement:**

4: Strong: This paper deepens the understanding of some phenomenon or lowers the barriers to an existing research direction.

**Paper Topic And Main Contributions:**

This paper works on improving spoken language understanding (SLU). More specifically, it aims at exploring the usage of code-switched sentences for improving SLU in the zero-shot cross-lingual setting. This paper proposes better generation techniques for code-switched sentences through better word replacement and novel token-level alignment strategies. Models trained using their proposed method SoGo show superior performance across nine languages on the MultiATIS++ benchmark.

**Reasons To Accept:**

- Novel and well-written approach: The usage of task-level saliency for finding substitution words and hidden-state-based word alignments are both used in a novel way for generating code-switched sentences. The methods are also very well-explained despite the short space of 4 pages.
- Generalizable approaches to other tasks: Their work can have a larger impact as code-switched sentences are used in various different tasks and pre-training as well. So, it might be of good interest to the larger community.
- Good experimentation and analysis: They correctly validate their results on three tasks of SLU and compare them with baseline models. They further provide a good ablation study along with many analyses. Their experimentation looks solid to me.

**Reasons To Reject:**

- Another task: Nitpicking, but it would be nice to see a small experiment if this code-switched sentence generation technique works for any other tasks as well. It would also make the technique more generalizable.
- Lack of qualitative analysis: Great analysis done throughout the paper. But I would like to see actual qualitative examples of how the proposed technique is different and better than previous baselines. Overall, it would build a stronger intuition for the reader.

**Reproducibility:**

4: Could mostly reproduce the results, but there may be some variation because of sample variance or minor variations in their interpretation of the protocol or method.

**Reviewer Confidence:**

3: Pretty sure, but there's a chance I missed something. Although I have a good feel for this area in general, I did not carefully check the paper's details, e.g., the math, experimental design, or novelty.

**Typos Grammar Style And Presentation Improvements:**

- The small font in figures: It’s understandable that smaller space means graphs need to be squeezed. But currently, the font is too small for readers to view and understand these graphs/figures.

- L281: Spelling error. gramma -> grammar

- Formatting of captions in Tables and Figures is not consistent. The caption is on the top of Table 1 while it’s on the bottom for the figures. Better to keep it at the bottom I think.

---

> ### Author Rebuttal · Authors · 2023-08-28
>
> **R1:**
> Thanks for your valuable suggestion! Following [1], we conduct additional experiments for **natural language inference**. We use XNLI [2], which covers 15 languages for natural language inference. We feed a pair of sentences directly into the mBERT encoder and a task-specific classification layer is used for classification. Models are evaluated by the classification accuracy (ACC), and the results are shown in the table below.
> | Models |  Avg.  |
> |:------ |:---------------------:|
> | CoSDA-XLM_based |            75.3          |
> |&nbsp;&nbsp;&nbsp;&nbsp; w/ SoGo | **77.4**
> | CoSDA-mBERT_based |         69.7         |
> | &nbsp;&nbsp;&nbsp;&nbsp; w/ SoGo | **71.9** |
>
> We will include the detailed experimental settings as well as the additional results in the final version.
>
> **R2:**
> Thank you for your insightful suggestion! Figure 3 serves as a case visualization to demonstrate grammatical coherence. Since grammar is difficult to quantify, we employ the token density in the embedding space for illustration. In the final version, we plan to include three additional examples to further illustrate the effectiveness of our method (Due to website restrictions, pictures cannot be uploaded). Due to time constraints, we will include three more case studies in the final version to visually demonstrate the effectiveness of our proposed method.
>
> **On Typos Grammar Style And Presentation Improvements:**
> Thank you for your careful reading! We will fix them in the final version.
>
> **References**
>
> [1] Qin L et al. Cosda-ml: Multi-lingual code-switching data augmentation for zero-shot cross-lingual nlp. IJCAI 2020
>
> [2] Conneau A et al. XNLI: Evaluating Cross-lingual Sentence Representations. EMNLP 2018

---

### Official Review · Reviewer_uvK4 · 2023-08-06

**Soundness:** 4

**Excitement:**

4: Strong: This paper deepens the understanding of some phenomenon or lowers the barriers to an existing research direction.

**Missing References:**


Cross-lingual Transfer Learning for Multilingual Task Oriented Dialog (Schuster et al 2019)

MTOP: A Comprehensive Multilingual Task-Oriented Semantic Parsing Benchmark (Li et al. 2021)


**Paper Topic And Main Contributions:**

This work presents a zero shot approach for spoken language understanding. Authors propose to train a model using code-switched sentences where the words to be code switched are chosen based on their saliency. In opposition to previous approaches (ie. Fazili and Jyothi, 2022; Whitehouse et al., 2022), where a bilingual dictionary is required, this approach scans all the vocabulary to identify a salient subset of keywords to generate code-switched sentences. To prevent changing the semantics of the resulting sentence, authors propose a Token-level Alignment that checks the similarity between hidden states of the original and the code switched sentence and considered as weight to be aggregated to the embedding of the original sentence.
The approach is checked on the MultiAtis++ dataset compared with other baselines such as  CoSDA (Qin et al., 2020) and GL-CLEF (Qin et al., 2022) as well as baseline mBERT, backbone of the presented model. Results show a consistent gain performance.
Ablation experiments are carried out	especially checking if the replacement based on saliency is bringing gains compared to replacing random words.


**Questions For The Authors:**

Is the algorithm applied solely on the training data from MultiAtis++?
If not, which data are you using?

**Reasons To Accept:**

Solid work, well justified, simple and straightforward.

Numbers hold.

Easy to reproduce.

Potentially applicable to other scenarios


**Reasons To Reject:**

The method is not completely clear

The method could be used on other datasets.


**Reproducibility:**

5: Could easily reproduce the results.

**Reviewer Confidence:**

4: Quite sure. I tried to check the important points carefully. It's unlikely, though conceivable, that I missed something that should affect my ratings.

---

> ### Author Rebuttal · Authors · 2023-08-28
>
> **R1:**
> Thank you for your valuable feedback! We will carefully review the method section and make necessary modifications in the final version to enhance clarity. Additionally, we plan to relocate the figure of the main framework to the main text, providing readers with a better understanding of the method.
>
> **R2:**
> Thank you for your valuable guidance! Drawing upon the literature and benchmarks you've provided, we conducted additional experiments on MTOP [2]. The results indicate that our proposed method SoGo exhibits robust generalization performance and obtains consistent improvements on the MTOP dataset. We will add these parts to the Appendix in the final version.
> | Intent Acc |  Avg.  |
> |:------ |:---------------------:|
> | CoSDA |            90.7          |
> |&nbsp;&nbsp;&nbsp;&nbsp; w/ SoGo | **92.4**
> | LAJ |         91.0         |
> | &nbsp;&nbsp;&nbsp;&nbsp; w/ SoGo | **93.3** |
> | **Sloft F1** |  **Avg.**  |
> | CoSDA |            73.3          |
> |&nbsp;&nbsp;&nbsp;&nbsp; w/ SoGo | **75.6**
> | LAJ |         74.5         |
> | &nbsp;&nbsp;&nbsp;&nbsp; w/ SoGo | **77.7** |
> | **Overall Acc** |  **Avg.**  |
> | CoSDA |            58.8          |
> |&nbsp;&nbsp;&nbsp;&nbsp; w/ SoGo | **61.5**
> | LAJ |        60.1         |
> | &nbsp;&nbsp;&nbsp;&nbsp; w/ SoGo | **64.8** |
>
> **Q1:**
> Yes, we conduct cross-lingual tasks in the zero-shot setting, in which only English labeled sentences with code-switching augmentation are used for training, and evaluation is performed in all other languages. Note that the English labeled sentences used for training are only from MixATIS++, and the code-switching augmentation does not extend beyond its training set.
>
> **On references:**
> Thank you for your valuable references! Due to time constraints, we will report the complete experimental results of these two datasets [1, 2] in the final version.
>
> **References**
>
> [1] Schuster S et al. Cross-lingual transfer learning for multilingual task oriented dialog. NAACL 2019
>
> [2] Li H et al. MTOP: A comprehensive multilingual task-oriented semantic parsing benchmark. arXiv 2020

---

### Official Review · Reviewer_YoWR · 2023-08-10

**Soundness:** 4

**Excitement:**

3: Ambivalent: It has merits (e.g., it reports state-of-the-art results, the idea is nice), but there are key weaknesses (e.g., it describes incremental work), and it can significantly benefit from another round of revision. However, I won't object to accepting it if my co-reviewers champion it.

**Missing References:**

N/A

**Paper Topic And Main Contributions:**

This paper proposes a novel framework, SoGo, for zero-shot cross-lingual spoken language understanding to improve the semantic and grammar coherence. Previous approaches using code-switched sentences have two main problems: (1) randomly replace the word with equal probability, however those words have different levels of importance to sentences. (2) lack grammatical coherence and token-level dependencies are disregarded.

The paper makes two main contributions to solve the problems: (1) use saliency-based substitution to choose the top-k words from sentences and gather all the salient words into a salient subset K. (2) after replace the salient words, use token-level alignment to match the code-switched sentence to the original sentence,  so as to improve the token-level dependencies and grammatical coherence. The experiment results show that those two methods both increase the accuracy of intent detections and slot filling compared with vanilla models.

**Questions For The Authors:**

Question A: in the saliency-based substitution part, why words with high document frequency has lower weight? Won't this mitigate the saliency of those words?

Question B: in the saliency-based substitution part, are the saliency scores normalized over a sentence before they were added?

Question C: It is hard to relate the figure 4(b) to model part in section 3, like what is Lsf and Lid in the figure. Shouldn't them be Ls and Li? There is no illustration in the figure caption or main paper

Question D: in the implementation part, why the random rate is set to 0.2? You just randomly chose 0.2?

Question E: in the table 1, in the Slot F1 section en column, the LAJ-MCL section  has the highest score, why SoGo score is bolded? Same thing in the overcall acc zh column.

Question F: in the analysis of saliency-based substitution part, do you use only one sentence to plot? Cause it is said "within a random sentence". It seems that the data points in the figure are quite little. Is the plot based on multiple sentences or only one? If it is based on multiple sentences, what is the number of sentences you used. If it is based on one sentence, isn't it arbitrary to has this conclusion? Figure 3 needs more illustration and analysis.

Question G: no need to answer, just a suggestion: put the full name of SoGo in the beginning, not in the conclusion

**Reasons To Accept:**

Main strengths: This paper proposed two solutions: saliency-based substitution to find important words for replacement in a sentence; token-level alignment to improve token-level dependencies and grammatical coherence. This paper discovered the possible problems in SLU, employed and modified existing techniques to improve the model. The results show that their new method performs better than previous models generally. In their ablation study, they also showed that either solution solely can improve the accuracy.

The main benefit of this paper is that both methods are easy to understand and reproduce. Also the results show that the methods are quite promising.

**Reasons To Reject:**

1. Saliency-based method and token-alignment are not new, and the method part lacks illustration (more details in 'Questions' part)
2. The author designed the novel method based on the observation of previous models lacking grammatical coherence and token-level dependencies. However, how to know that the accuracy improvement has direct relation to those, especially the grammatical coherence. The token level alignment might improve grammatical coherence and token-level dependencies but not necessary. It would be better if authors studied some examples from vanilla models output and new model output, so as to see if the accuracy are really improved because of coherence and dependencies.
3. The improvement is not significant. Compared to GL-CLEF and LAJ-MCL (other two models mentioned in the table), almost half the results (14 out of 30) have less than 0.5% accuracy/F1 increment.

**Reproducibility:**

4: Could mostly reproduce the results, but there may be some variation because of sample variance or minor variations in their interpretation of the protocol or method.

**Reviewer Confidence:**

4: Quite sure. I tried to check the important points carefully. It's unlikely, though conceivable, that I missed something that should affect my ratings.

**Typos Grammar Style And Presentation Improvements:**

line 137 denotes->denote
line 123 are optimized -> is optimized

---

> ### Author Rebuttal · Authors · 2023-08-28
>
> **R1:**
> Thank you for your professional review. Saliency methods are commonly employed for model interpretability, while weighted token alignment has also been explored in tasks like NER and POS tagging. However, there remains a gap in related literature on cross-lingual dialogue. Our aim is to intricately devise two techniques for enhancing code-switch augmentation to address two key issues in dialogue understanding tasks, with the intention of extending their applicability to a broader spectrum of dialogue scenarios, including Dialogue State Tracking and Sentiment Classification, etc. If you are interested, please see Reviewer *#nHnV* for preliminary experiments on more tasks like NLI.
>
>
> **R2:**
> Thank you for your valuable suggestion! Figure 3 serves as a case visualization to demonstrate grammatical coherence. Since grammar is difficult to quantify, we employ the token density in the embedding space for illustration. In the final version, we plan to include three additional examples to further illustrate the effectiveness of our method (*Due to website restrictions, pictures cannot be uploaded*).
>
>
> **R3:**
> We would like to respectfully clarify the results and substance of our paper. To reduce the influence of the randomness caused by the token substitution, all results reported in the paper are averaged by 5 runs. Regarding the main table, we have restructured it to enhance clarity. The results show that our proposed SoGo achieves significant and consistent performance gains over the two baseline models.
>
> |  **Intent Acc**  |  **Avg.**   |
> |:------ |:---------------------:|
> | CoSDA | 87.32 |
> |&nbsp;&nbsp;&nbsp;&nbsp; w/ SoGo | **90.56** ($\uparrow$3.24)  |
> | GL-CLeF |         91.95         |
> | &nbsp;&nbsp;&nbsp;&nbsp; w/ SoGo | **92.69** ($\uparrow$0.74) |
> | **Sloft F1** |  **Avg.**  |
> | CoSDA |            73.47          |
> |&nbsp;&nbsp;&nbsp;&nbsp; w/ SoGo | **76.14** ($\uparrow$2.67)
> | GL-CLeF |         80.00         |
> | &nbsp;&nbsp;&nbsp;&nbsp; w/ SoGo | **81.64** ($\uparrow$1.64) |
> | **Overall Acc** |  **Avg.**  |
> | CoSDA |            44.03          |
> |&nbsp;&nbsp;&nbsp;&nbsp; w/ SoGo | **49.08** ($\uparrow$5.05)
> | GL-CLeF |        52.50         |
> | &nbsp;&nbsp;&nbsp;&nbsp; w/ SoGo | **57.02** ($\uparrow$4.52) |
>
>
> **QA:**
> The IDF term balances word frequency and saliency scores by assigning words with high document frequency a lower weight and vice versa. It is necessary because some irrelevant stop words (e.g., "of" and "a") have high total saliency scores, for they appear in the sentence many times.
>
> **QB:**
> Thank you for your thorough consideration! Yes, the saliency scores are normalized to ensure training stability.
>
>
> **QC:**
> Thank you for your attentive review! Yes, Lsf and Lid in Figure 4 correspond to Ls and Li in Equation 8. We apologize for the inconvenience caused by the figure version update issue. In the final version, we will make the necessary adjustments to relocate Figure 4 to the main text, thus enhancing the overall clarity of the method.
>
>
> **QD:**
> Following [1, 2], the random rate for token substitution is set to 0.2 for a fair comparison.
>
> **QE:**
> Apologies for the mistakes! We will correct the incorrect bold formatting in the final version.
>
> **QF:**
> Thank you for this concern! Note that Figure 3 has been indeed visualized for a single sentence. We observe that our model achieves a more cohesive space for token sequences, where the switched tokens are well-integrated within the context. In contrast, the previous CoSDA method separates the substituted tokens from the original context, resulting in clear distinctions between tokens from different languages. Our approach maintains a consistent embedding space throughout training, fostering effective cross-lingual representations. In the final version, we plan to include three additional examples to further illustrate the effectiveness of our method (*Due to website restrictions, pictures cannot be uploaded*).
>
> **On typo and presentation suggestion:**
> Thank you for your careful reading! We will fix them in the final version.
>
> **References**
>
> [1] Qin L et al. GL-CLeF: A Global-Local Contrastive Learning Framework for Cross-lingual Spoken Language Understanding. ACL 2022
>
> [2] Qin L et al. Cosda-ml: Multi-lingual code-switching data augmentation for zero-shot cross-lingual nlp. IJCAI 2020
>
> We hope that you may consider reevaluating the paper in light of these clarifications and additional experiments that we have performed.

---

### Official Review · Reviewer_ncFV · 2023-08-11

**Typos Grammar Style And Presentation Improvements:** X
**Soundness:** 4

**Excitement:**

2: Mediocre: This paper makes marginal contributions (vs non-contemporaneous work), so I would rather not see it in the conference.

**Missing References:**

- Y. Feng, F. Li, and P. Koehn, “Toward the Limitation of Code-Switching in Cross-Lingual Transfer,” in Proceedings of the 2022 Conference on Empirical Methods in Natural Language Processing, Abu Dhabi, United Arab Emirates: Association for Computational Linguistics, Feb. 2022, pp. 5966–5971. Accessed: Feb. 20, 2023. [Online]. Available: https://aclanthology.org/2022.emnlp-main.400

**Paper Topic And Main Contributions:**

This paper proposes a better code-switching method for SLU methods. The authors propose how to select words critical for code-switching, and how to consider the contextual information (semantics and grammar).

**Questions For The Authors:**

- How should we decide the hyperparameter of replacement rate? As depicted in Fig 2, the performance fairly depends on the replacement ratio.

**Reasons To Accept:**

- The proposed method outperforms existing methods
- The ablation studies shows the effectiveness of each proposal. PCA visualization is intriguing also.

**Reasons To Reject:**

[Half of proposal is same with existing work]
- The second proposal (token-level alignment) is exactly the same with the method from [Feng et al.](https://aclanthology.org/2022.emnlp-main.400/), but never mentioned in the manuscript.

[More evaluation needed]
- Since the method depends on the quality of dictionaries, the method may not be efficiently applied to low-resourced languages, which is the goal of the paper (line 3,34). What if the quality of dictionary is low? For example, if the number of word pairs is low, even if we select top words based on saliency, we may not be able to substitute the words. Using unsupervised dictionary generation (e.g. fast_align) could be more supportive
- The principle of Saliency-based Substitution is basically same with [Liu et al.](https://ojs.aaai.org/index.php/AAAI/article/view/6362), who selects the top-k important words based on the attention values. Comparing the selection method must make this submission stronger.

**Reproducibility:**

2: Would be hard pressed to reproduce the results. The contribution depends on data that are simply not available outside the author's institution or consortium; not enough details are provided.

**Reviewer Confidence:**

4: Quite sure. I tried to check the important points carefully. It's unlikely, though conceivable, that I missed something that should affect my ratings.

---

> ### Author Rebuttal · Authors · 2023-08-29
>
> **R1:**
> Thank you for your professional review! Saliency methods are commonly employed for model interpretability, while weighted token alignment has also been explored in tasks like NER and POS tagging [1]. However, there remains a gap in related literature on cross-lingual dialogue. Our aim is to intricately devise two techniques for enhancing code-switch augmentation to address two key issues in dialogue understanding tasks, with the intention of extending their applicability to a broader spectrum of dialogue scenarios, including Dialogue State Tracking and Sentiment Classification, etc. If you are interested, please see Reviewer #nHnV for preliminary experiments on more tasks like NLI. **We promise to include the missing related references to the best of our knowledge, and further clarify the connection and difference in the final version**.
>
> **R2:**
> Thanks for your valuable suggestions! As indicated in Line #298-304 in Limitation, we **have pointed out** the limitations of the proposed method that rely on dictionaries. This limitation is inherent in all similar code-switching methods, and it is not specific to the technique employed in our proposed method. This is not the focus of this work.
> The suggestion of employing unsupervised dictionary generation, such as fast_align, is valuable and will be considered in our future work.
>
> **R3:**
> Thank you for your insightful references! As indicated in Line #69-71, [2] computes attention scores for pairs of source and target languages individually, leading to different models for each target language. In contrast, our proposed method employs a uniform saliency score to map multiple languages into a consistent space, and thus only requires a single training process for multiple target languages. To thoroughly address your concerns, we have implemented adjustments to the proposed method, replacing the saliency component with an attention-based approach. The additional results on CoSDA are presented in the Table below (averaged by 3 runs).
> | Overall Acc |  Avg.  |
> |:------ |:---------------------:|
> | Attention-based |            47.86          |
> | Saliency-based | 49.08 |
>
> Based on the outcomes, we intuitively infer that the saliency-based method offers improved interpretability for code word replacement, as it can effectively decide the more salient words in the dialogue context.
>
> Here, we want to further discuss the differences between the two: Since attention mechanisms boosted performance on many current NLP tasks [3], using attention weight as the explanation of model predictions is a general approach for many models [4, 5]. However, some recent works [6, 7] cast doubt on attention's interpretability. Besides, [8] claimed that saliency methods are more applicable to model explanations. Therefore, we propose saliency-based substitution for cross-lingual SLU task.
>
> As you suggested, we consider the discussion of this aspect to be crucial, and we will incorporate it into the main text in the final version to clarify the effectiveness of our proposed method.
>
> **Q1:**
> Thank you for your concern! In order to provide further validation for the model's generalizability, we conducted additional experiments on MTOP [9]. These results also show notable and consistent improvements with our proposed method.
>
> | Intent Acc |  Avg.  |
> |:------ |:---------------------:|
> | CoSDA |            90.72          |
> |&nbsp;&nbsp;&nbsp;&nbsp; w/ SoGo | **92.48**
> | LAJ |         91.04         |
> | &nbsp;&nbsp;&nbsp;&nbsp; w/ SoGo | **93.30** |
> | **Sloft F1** |  **Avg.**  |
> | CoSDA |            73.34          |
> |&nbsp;&nbsp;&nbsp;&nbsp; w/ SoGo | **75.63**
> | LAJ |         74.50         |
> | &nbsp;&nbsp;&nbsp;&nbsp; w/ SoGo | **77.74** |
> | **Overall Acc** |  **Avg.**  |
> | CoSDA |            58.77          |
> |&nbsp;&nbsp;&nbsp;&nbsp; w/ SoGo | **61.52**
> | LAJ |        60.11         |
> | &nbsp;&nbsp;&nbsp;&nbsp; w/ SoGo | **64.80** |
>
> Due to time constraints, we conducted tests using replacement rates of [20%, 30%, 40%]. Interestingly, MTOP exhibited a similar performance trend to MixATIS++. Therefore, we empirically conclude that an approximately 40% replacement rate serves as the optimal hyper-parameter in most scenarios. We will report the complete hyper-parameter experiment results in the final version.
>
> **On reproducibility:**
> Following the common practice, the source code of this paper will be released upon publication.
>
> **References**
>
> [1] Feng Y et al. Toward the Limitation of Code-Switching in Cross-Lingual Transfer. EMNLP 2022
>
> [2] Liu Z et al. Attention-informed mixed-language training for zero-shot cross-lingual task-oriented dialogue systems. AAAI 2020
>
> [3] Bahdanau D et al. Neural machine translation by jointly learning to align and translate. arXiv 2014
>
> [4] Lin Z et al. A structured self-attentive sentence embedding. arXiv 2017
>
> [5] Ghaeini R et al. Interpreting Recurrent and Attention-Based Neural Models: a Case Study on Natural Language Inference. EMNLP 2018
>
> [6] Serrano S et al. Is Attention Interpretable? ACL 2019
>
> [7] Wiegreffe S et al. Attention is not not Explanation. EMNLP 2019
>
> [8] Bastings J et al. The elephant in the interpretability room: Why use attention as explanation when we have saliency methods? Third BlackboxNLP Workshop on Analyzing and Interpreting Neural Networks for NLP  2020
>
> [9] Li H et al. MTOP: A comprehensive multilingual task-oriented semantic parsing benchmark. arXiv 2020
>
> We appreciate if you would consider increasing your score in light of our response.

---

### Meta-Review · Area_Chair_kjfQ · 2023-09-19

**Recommendation:** 4

**Metareview:**

Based on the reviews, the paper proposed a better method for code-switching in spoken language understanding (SLU). The main contributions include the selection of critical words for code-switching and the consideration of contextual information. Reviewers appreciated the outperformance of the proposed method compared to existing methods and the effectiveness shown in the ablation studies. However, there were concerns about the lack of novelty in one of the proposals and the dependency on the quality of dictionaries, which may limit the applicability to low-resourced languages.

On the positive side, the reviewers mentioned that the paper is easy to understand and reproduce, and the results are promising. They also found the proposed techniques to be well-explained and the experimentation and analysis to be solid. However, they suggested conducting more qualitative analysis to provide concrete examples of the improvements. They highlighted the potential generalizability of the proposed techniques to other tasks.

To summarize, the work is considered to be of strong soundness and it provides sufficient support for its claims and arguments. There is generally strong excitement about the paper, as it contributes to the understanding of code-switching in SLU and offers potential improvements. However, there are concerns about the lack of novelty in one aspect of the work and the limited applicability to low-resourced languages. The reviewers suggest including more qualitative analysis and exploring the generalizability of the proposed techniques.

---

### Decision · Program_Chairs · 2023-10-07

**Decision:**

Accept-Main

**Comment:**

Based on the reviews, the paper proposed a better method for code-switching in spoken language understanding (SLU). The main contributions include the selection of critical words for code-switching and the consideration of contextual information. Reviewers appreciated the outperformance of the proposed method compared to existing methods and the effectiveness shown in the ablation studies. However, there were concerns about the lack of novelty in one of the proposals and the dependency on the quality of dictionaries, which may limit the applicability to low-resourced languages.

On the positive side, the reviewers mentioned that the paper is easy to understand and reproduce, and the results are promising. They also found the proposed techniques to be well-explained and the experimentation and analysis to be solid. However, they suggested conducting more qualitative analysis to provide concrete examples of the improvements. They highlighted the potential generalizability of the proposed techniques to other tasks.

To summarize, the work is considered to be of strong soundness and it provides sufficient support for its claims and arguments. There is generally strong excitement about the paper, as it contributes to the understanding of code-switching in SLU and offers potential improvements. However, there are concerns about the lack of novelty in one aspect of the work and the limited applicability to low-resourced languages. The reviewers suggest including more qualitative analysis and exploring the generalizability of the proposed techniques.